# Diagnostic Approaches for *Candida auris*: A Comprehensive Review of Screening, Identification, and Susceptibility Testing

**DOI:** 10.3390/microorganisms13071461

**Published:** 2025-06-24

**Authors:** Christine Hsu, Mohamed Yassin

**Affiliations:** 1College of Medicine, National Cheng Kung University, Tainan 701401, Taiwan; christine050122@gmail.com; 2Monmouth Medical Center, Robert Wood Johnson/Barnabas Health Care System, Long Branch, NJ 07740, USA; 3Infectious Diseases and Infection Prevention Department, School of Medicine and Public Health Pittsburgh, University of Pittsburgh, Pittsburgh, PA 15219, USA

**Keywords:** *Candida auris*, diagnostic methods, matrix-assisted laser desorption/ionization time-of-flight mass spectrometry (MALDI-TOF MS)

## Abstract

*Candida auris* (*C. auris*) is an emerging multidrug-resistant fungal pathogen recognized by the World Health Organization (WHO) as a critical global health threat. Its rapid transmission, high mortality rate, and frequent misidentification in clinical laboratories present significant challenges for diagnosis and infection control. This review provides a comprehensive overview of current and emerging diagnostic methods for *C. auris* detection, including culture-based techniques, biochemical assays, matrix-assisted laser desorption/ionization time-of-flight mass spectrometry (MALDI-TOF MS), and molecular diagnostics such as PCR and loop-mediated isothermal amplification (LAMP). We evaluate each method’s sensitivity, specificity, turnaround time, and feasibility in clinical and surveillance settings. While culture remains the diagnostic gold standard, it is limited by slow turnaround and phenotypic overlap with related species. Updated biochemical platforms and MALDI-TOF MS with expanded databases have improved identification accuracy. Molecular assays offer rapid, culture-independent detection. Antifungal susceptibility testing (AFST), primarily using broth microdilution, is essential for guiding treatment, although standardized breakpoints remain lacking. This review proposes an integrated diagnostic workflow and discusses key innovations and gaps in current practice. Our findings aim to support clinicians, microbiologists, and public health professionals in improving early detection, containment, and management of *C. auris* infections.

## 1. Introduction

*Candida auris* (*C. auris*) has emerged as a significant global health threat due to its rapid spread, multidrug resistance, and high associated mortality. Recognized by the World Health Organization (WHO) as a critical priority pathogen in its inaugural fungal priority pathogen list, *C. auris* poses a serious challenge to public health worldwide [1]. Since its discovery in 2009, this opportunistic fungal pathogen has been reported across six continents, with a sharp rise in case numbers in recent years. In the United States, for example, the incidence of *C. auris* infections increased by 95% from 2019 to 2021, with new cases identified in 17 additional states during this period [2,3]. Similarly, in Europe, reported cases nearly doubled between 2020 and 2021 [4,5].

Distinct from other *Candida* species that primarily colonize the gastrointestinal tract, *C. auris* preferentially colonizes the skin. This enables efficient transmission in hospitals and long-term care facilities. Its ability to form biofilms, persist on surfaces, and resist standard disinfection protocols contributes to its capacity to cause widespread outbreaks [6]. Colonization also significantly elevates the risk of invasive infection. In certain healthcare settings in South Africa and India, *C. auris* has accounted for up to 25% and 40% of candidemia cases, respectively [2,7]. Global mortality rates linked to *C. auris* infections range between 40% and 60%, although attributing deaths solely to *C. auris* is often complicated by patients’ underlying comorbidities [8,9,10,11].

Effective infection control and prevention hinge on early and accurate detection. However, this remains a major challenge. Conventional diagnostic tools often misidentify *C. auris*, particularly conventional culture and biochemical assays, which frequently confuse it with closely related species such as *Candida haemulonii*, leading to diagnostic delays and inappropriate treatments [12,13]. Earlier iterations of MALDI-TOF MS also failed to correctly identify *C. auris* due to inadequate reference spectra [14]. Additionally, *C. auris* is genetically diverse, comprising at least six distinct clades, further complicating species identification and necessitating advanced molecular and proteomic techniques for precise diagnosis and epidemiological surveillance.

This review examines current diagnostic strategies for *C. auris*, encompassing both culture-dependent and culture-independent approaches. It evaluates the primary diagnostic modalities used in clinical and surveillance settings, with a focus on sensitivity, specificity, turnaround time, and operational feasibility. We highlight recent advancements that improve diagnostic accuracy and efficiency, assess their practical applications in laboratory and clinical contexts, and discuss key limitations. By providing an up-to-date and comprehensive overview, this review aims to guide clinicians, microbiologists, and public health professionals in enhancing early detection, outbreak management, and patient care outcomes.

## 2. Clinical Diagnostic Approaches of *C. auris*

The diagnosis of *C. auris* infection or colonization relies on both clinical and surveillance samples. Clinical cases refer to cultures collected for diagnosing or treating disease, whereas screening cases involve samples obtained for surveillance purposes. Accurate and timely identification is critical for guiding appropriate treatment, implementing effective infection control measures, and preventing healthcare-associated outbreaks. To assist with method selection across varied settings, Table 1 summarizes major diagnostic techniques for *C. auris*, comparing their performance characteristics and operational feasibility.

An integrated diagnostic workflow follows a multi-step process—from specimen collection to definitive identification—utilizing both culture-based and molecular technologies. As illustrated in Figure 1, this begins with presumptive identification using traditional culture on chromogenic media, followed by confirmatory testing via biochemical assays, MALDI-TOF MS, or molecular techniques such as PCR and LAMP.

Routine microbiological cultures remain the cornerstone of clinical diagnosis and are conducted on both sterile (e.g., blood) and non-sterile (e.g., urine, sputum, etc.) specimens. However, optimal growth of *C. auris* requires incubation at elevated temperatures (37–40 °C), a feature that distinguishes it from other *Candida* species. Without this thermotolerant condition, standard culture protocols may fail to detect the organism. Complete identification of yeast isolates from sterile sites to exclude *C. auris* is recommended [31]. Detection of *C. auris* in sterile specimens such as blood is clinically significant and indicates invasive candidiasis. Importantly, *Candida*-positive blood cultures should never be dismissed as contaminants.

In contrast, diagnosing *C. auris* from non-sterile specimens presents unique challenges. Sites like the respiratory or urinary tract often yield mixed yeast flora, necessitating additional tests to confirm or exclude *C. auris*. This complexity increases laboratory workload and may delay reporting, particularly in settings with limited resources [32]. Therefore, balancing comprehensive identification with practical laboratory constraints is essential.

To address these limitations, molecular diagnostics such as PCR have been developed. PCR offers high sensitivity and specificity for both clinical and surveillance samples and delivers results significantly faster than traditional culture methods. However, its use is typically reserved for cases with a high index of suspicion, owing to cost and workflow constraints. In parallel, supplementary assays like beta-D-glucan (BDG) and mannan/anti-mannan antibody detection are available, but they have limited diagnostic utility due to their low specificity and sensitivity for invasive candidiasis—particularly in cases involving *Candida auris* and *Candida parapsilosis* [29,30,33,34].

Ultimately, clinical judgment plays a vital role in interpreting the significance of *C. auris* isolation. Detection from non-sterile sites may reflect colonization rather than true infection. Nonetheless, even in such cases, identifying colonized individuals is essential, particularly in outbreak settings, as it supports targeted infection prevention and control measures [31].

## 3. Screening Guidelines and Protocols of *C. auris*

### 3.1. Importance of Screening for C. auris Colonization

Screening for *C. auris* colonization plays a pivotal role in early detection and infection control. This pathogen spreads readily through contact with high-touch surfaces, contaminated medical equipment, and colonized patients. Its ability to persist on surfaces for 7 to 14 days facilitates continued environmental contamination and nosocomial transmission. Documented cases show that some patients can acquire *C. auris* infections within just four days of exposure in healthcare settings [35,36,37,38,39].

Colonization is often prolonged, particularly in intensive care units (ICUs), long-term care environments, and regions where *C. auris* is endemic [40]. Patients may remain colonized throughout their hospital stay, and even those discharged into the community may carry the fungus for up to eight months [41,42,43]. Although colonization is typically asymptomatic, individuals continue to shed the organism, making them ongoing sources of transmission and elevating the risk of invasive infection.

Progression from colonization to infection is uncommon but significant. One study estimated 0.3 invasive cases per 1000 patient-days among colonized individuals. [44]. Reported mortality rates for *C. auris* infections range between 40% and 60%, although the exact contribution is difficult to quantify due to underlying comorbidities [8,9]. Given the severity of potential outcomes, proactive screening is essential for limiting outbreaks. Healthcare facilities may conduct screening either as a preventive measure, targeting high-risk individuals before transmission occurs, or as a response-based intervention following the identification of a confirmed case to assess secondary spread and infection control efficacy.

### 3.2. Identifying Individuals for Screening

Effective screening strategies begin with identifying populations at elevated risk. High-priority individuals include those with epidemiologic links to known cases, recent exposure to healthcare environments, or clinical factors that predispose them to colonization [45]. Environmental screening is also important, but this study focuses on human screening.

Patients transferred from long-term acute care hospitals (LTACHs) or ventilator-capable skilled nursing facilities (vSNFs) are particularly vulnerable [41]. Prolonged hospital stays, especially in ICUs, are also associated with increased colonization rates and a higher risk of invasive candidiasis [46]. Even after a recent negative screen, repeat testing may be warranted upon ICU discharge [40].

Those who have shared rooms or equipment with confirmed *C. auris* cases or received care from the same healthcare personnel during overlapping timeframes also face heightened risks. Studies have found colonization rates of 5% to 10% among close contacts [47]. Despite this, routine screening of healthcare workers or household members is not recommended unless specific risk factors are present [31].

The necessity of admission screening varies by region [48]. In non-endemic areas, community prevalence is low, and routine screening upon hospital admission is not standard practice. Studies from the UK and USA support this, showing that admission carriage is rare and usually associated with prior hospital exposure [49,50]. In contrast, endemic countries like India report colonization rates as high as 9.3% on admission, supporting the use of targeted screening protocols upon hospital entry in these contexts [51]. Individuals with prior healthcare exposure in these regions should undergo screening upon hospital admission [47,52].

Beyond epidemiological exposure, several clinical risk factors further elevate susceptibility. These include mechanical ventilation, use of central venous catheters, parenteral nutrition, prior antifungal or broad-spectrum antibiotic use, recent surgery, and episodes of sepsis. Patients with immunosuppressive conditions—such as HIV infection, malignancy, neutropenia, corticosteroid therapy, or organ transplantation—are also at greater risk. Additional comorbidities like liver cirrhosis, renal disease, diabetes mellitus, and history of blood transfusions have been linked to increased colonization and infection rates [1,53,54,55]. Screening decisions should ultimately be guided by local epidemiologic data, facility-specific transmission patterns, and individual patient risk profiles.

### 3.3. Screening Methods, Timing, and Laboratory Protocols

Once at-risk individuals are identified, appropriate screening methods must be selected to ensure diagnostic accuracy and timely intervention. The CDC recommends using a composite swab of the axilla and groin for colonization screening, as these areas consistently exhibit the highest fungal burden and first-detection positivity [56,57] (Figure 2). Some studies suggest that including nasal swabs alongside axilla and groin may improve detection rates further [58]. In contrast, other sites such as urine, throat, pharynx, and axillary-rectal swabs have demonstrated lower sensitivity and are not considered reliable for routine screening [56,59]. Nonetheless, *C. auris* may colonize additional body surfaces that are not regularly sampled, allowing undetected transmission to persist in clinical settings [59].

A standard protocol involves using a nylon-flocked swab (e.g., ESwab) to firmly sample both axillae and the groin. The swab is then placed in a transport medium [60], labeled accurately, sealed in a biohazard bag, and submitted to the laboratory for processing. To avoid false-negative results, screening should be postponed for at least 48 h if the patient has recently received topical antiseptics [13].

The optimal laboratory method for detecting *C. auris* colonization remains under investigation. However, the CDC currently endorses several approaches, including culture on chromogenic or salt/dulcitol agar, molecular detection via PCR, and salt/dulcitol enrichment broth. Each method varies in sensitivity, turnaround time, and resource requirements.

A positive screening result confirms colonization, indicating the presence of *C. auris* on the patient’s skin even in the absence of symptoms. Because colonized individuals are capable of transmitting the pathogen, a confirmed result necessitates the immediate implementation of transmission-based precautions. Additionally, healthcare facilities are required to report cases to the CDC and notify appropriate local or state health departments to support regional infection control measures.

## 4. Diagnostic Methods for *C. auris* Detection

### 4.1. Culture-Based Methods

#### 4.1.1. Differential and Selective Media

Culture-based diagnostics remain the gold standard for the detection of *C. auris* in both clinical and surveillance samples. These methods are particularly important because advanced identification techniques such as biochemical assimilation tests and MALDI-TOF MS require isolated yeast colonies as input. However, conventional culture methods alone are often insufficient for accurate identification, as *C. auris* shares significant phenotypic similarity with other *Candida* species—most notably *C. haemulonii* and *C. duobushaemulonii*—making morphological differentiation unreliable [61,62,63,64].

Phenotypic variability in response to temperature and salinity further complicates the identification process. The CDC has cautioned against relying solely on colony morphology for screening purposes, emphasizing the need for more selective approaches [65,66].

To address these limitations, selective media have been developed by exploiting the unique thermotolerance, halotolerance, and carbohydrate assimilation profile of *C. auris*. The fungus grows on standard media such as Sabouraud dextrose agar (SDA) and various chromogenic formulations at optimal temperatures of 37–40 °C but shows reduced growth at 42 °C—an important differentiator from related species [67]. Its ability to tolerate saline environments, including concentrations of 10–12.5% NaCl, has been used in selective formulations to suppress competing flora [58,68]. Additionally, unlike *C. haemulonii*, which depends on glucose, *C. auris* can metabolize dulcitol and mannitol [6,36].

The first targeted selective medium was developed by Welsh et al. (2017), incorporating 10% NaCl and elevated temperatures (40 °C) into Sabouraud or yeast nitrogen base broth with dulcitol or mannitol as the carbon source [36]. Since then, newer non-commercial formulations have emerged. The selective auris medium, a modified yeast extract peptone dextrose agar (YPD agar) with 12.5% NaCl and 9 mM ferrous sulfate, allows direct plating from blood cultures at 42 °C [15]. Similarly, the specific *C. auris* medium (SCA), developed by Walsh et al., contains crystal violet to inhibit *Candida tropicalis* and has shown promising results in ICU settings in Algeria [16,69]. Despite these advances, additional validation against closely related species in the *C. haemulonii* complex remains necessary to ensure specificity.

In resource-limited settings, simpler media can be supplemented with fluconazole to exploit *C. auris*’ intrinsic triazole resistance or with chloramphenicol and gentamycin to suppress bacterial and fungal contaminants [36,70]. These adaptations offer cost-effective alternatives for presumptive screening in endemic regions.

#### 4.1.2. Chromogenic Media

Chromogenic media are frequently used for presumptive identification due to their accessibility, affordability, and ease of use. These media enable visual differentiation of yeast species based on colony color (Figure 3), typically within five days. Specimens can be applied directly or after broth enrichment, which, although more labor-intensive, improves detection sensitivity by one log_10_ CFU/mL [71]. However, colony color alone is not a reliable identifier; *C. auris* colonies may appear cream, pink, red, or purple—colors shared with other *Candida* species—thus complicating interpretation [20].

Recent formulations have aimed to address this limitation. CHROMagar™ Candida Plus (CHROMagar, France) produces a distinctive light-blue colony with a blue halo for all four known *C. auris* lineages after 36–48 h of incubation at 37 °C, effectively differentiating it from most other *Candida* species, including members of the *C. haemulonii* complex. Despite high sensitivity and specificity, some false positives have been reported with species such as *Candida diddensiae*, *C. metapsilosis*, and *C. orthopsilosis* [18,72,73].

Other formulations, such as Pal’s medium used in combination with CHROMagar Candida, have demonstrated promise in distinguishing *C. auris* from similar species, though broader validation is still required [17]. HiCrome™ *C. auris* medium offers results within 24 h but has demonstrated variable accuracy across studies [20,70].

Temperature and media supplementation can enhance performance. Although manufacturers typically recommend 37 °C incubation, raising the temperature to 40 °C can suppress the growth of competing species (CDC). Additionally, the use of fluconazole-supplemented CHROMagar improves specificity by inhibiting susceptible strains [57,64].

#### 4.1.3. Conclusion of Culture-Based Methods

Despite the high risk of misidentification and relatively long turnaround times—typically 24 to 48 h and up to 10 days in some cases [74]—culture-based methods, including both traditional and selective approaches, remain essential components of the diagnostic pathway. These methods serve as the foundation for subsequent analyses such as biochemical identification, MALDI-TOF MS, and AFST. Given the potential for diagnostic errors, all presumptive positive cultures should be confirmed using high-resolution diagnostic tools to ensure accurate species identification and inform appropriate clinical management.

### 4.2. Biochemical Assimilation Tests

Biochemical assimilation tests identify *C. auris* based on the organism’s metabolic profile, evaluating carbohydrate and nitrogen assimilation as well as enzymatic activity. A standardized yeast suspension is inoculated into wells containing various biochemical substrates. These panels are incubated, and identification is made by observing metabolic-induced color changes or turbidity, which are then compared against a reference database [75].

Despite their widespread use, these methods frequently misidentify *C. auris* due to significant overlap in metabolic profiles with other *Candida* species. Rather than yielding “no identification”, they often produce incorrect matches, particularly with species from the *C. haemulonii* complex.

Historically, early *C. auris* isolates were commonly misidentified by biochemical platforms. For example, the API 20C AUX and API ID 32C systems often reported isolates as *Rhodotorula glutinis*, *Saccharomyces cerevisiae*, or *Candida sake*, while the VITEK 2 system misclassified them as *C. haemulonii* [76,77,78,79]. Subsequent updates to the VITEK 2 system, particularly version 8.01, improved the accuracy of identifying isolates belonging to the South American Clade (Clade IV). However, identification remains inconsistent for isolates from the South Asian (Clade I) and African (Clade III) clades [80,81]. Misidentifications continue to occur, with isolates reported as *C. haemulonii*, *C. duobushaemulonii*, or even as unidentified *Candida* species [77]. Although VITEK 2 version 9.01 has been released, its performance in identifying a broader range of clades has not yet been fully evaluated.

Other commonly used biochemical platforms, such as MicroScan WalkAway and BD Phoenix, currently lack database support for *C. auris*, resulting in frequent misclassification or failed identification [75,76]. Given these limitations, the CDC recommends that biochemical test results always be interpreted cautiously and in conjunction with confirmatory diagnostics, such as MALDI-TOF MS or DNA sequencing, as outlined in their 2019 algorithm (Table 2) [21].

In resource-limited settings, where MALDI-TOF MS and molecular diagnostics may not be available, biochemical assimilation tests remain a practical and cost-effective alternative. Among available platforms, the VITEK 2 system (version 8.01) has demonstrated relatively reliable performance for identifying *C. auris* and may also support rapid antifungal susceptibility testing, as discussed in the subsequent susceptibility testing section.

### 4.3. Matrix-Assisted Laser Desorption/Ionization Time-of-Flight Mass Spectrometry (MALDI-TOF MS)

#### 4.3.1. The Evolution of MALDI-TOF on Detecting *C. auris*

Since its introduction in the late 1990s, matrix-assisted laser desorption/ionization time-of-flight mass spectrometry (MALDI-TOF MS) has transformed clinical microbiology due to its speed, cost effectiveness, and high diagnostic accuracy [82]. Initially employed for bacterial identification, the platform was expanded to include yeasts such as *Candida* spp. by 2009 [83,84]. The subsequent inclusion of *C. auris* spectra in commercial and curated databases has established MALDI-TOF MS as a powerful tool for the rapid and reliable identification of this emerging pathogen.

This technique is culture-dependent and typically uses colonies from agar plates, enrichment broths, or blood culture bottles. The process works by ionizing microbial proteins to generate a unique mass-to-charge ratio spectrum, which is compared against reference databases (Figure 4). MALDI-TOF MS offers over 90% identification accuracy and is particularly effective in distinguishing *C. auris* from closely related species within the *C. parapsilosis* complex (*C. parapsilosis*, *C. orthopsilosis*, and *C. metapsilosis*), where conventional biochemical and culture-based methods often fall short [85,86,87].

In addition to its high precision, MALDI-TOF MS delivers results rapidly—typically within 4 to 5 h following culture growth—making it a critical tool for timely diagnosis and confirmation of *C. auris* infections in clinical settings [87]. Its utility depends heavily on the quality of reference libraries and appropriate sample preparation.

#### 4.3.2. Sample Preparation and Database Considerations

The diagnostic performance of MALDI-TOF MS is influenced by multiple factors, including extraction methods, database versions, and instrument calibration. Common sample preparation protocols include direct smear, on-plate extraction, full tube extraction, and partial tube extraction. Among these, full tube and partial tube extraction methods generally yield superior confidence scores for *C. auris* identification compared to direct smear or on-plate extraction [88,89].

Recent studies recommend partial tube extraction for its balance of diagnostic accuracy and workflow efficiency [89]. On-plate extraction, although slightly less sensitive, remains a viable cost- and time-effective method for settings with limited resources or during high-throughput testing, such as in bloodstream infection management [90].

The effectiveness of MALDI-TOF MS also depends on the version and comprehensiveness of its reference database. Major platforms such as the VITEK MS (bioMérieux) and Microflex LT Biotyper (Bruker Daltonics) have improved *C. auris* identification with updated databases like bioMérieux’s IVD v3.2 or RUO Saramis v4.14, and Bruker’s CA System (version 4) or RUO 2014 (5627). However, older library versions may still misidentify *C. auris* as *C. haemulonii* or *C. lusitaniae*, emphasizing the need for continuous database updates and confirmatory testing when results are ambiguous.

To enhance diagnostic reliability, laboratories can supplement manufacturer databases with free resources such as the CDC’s MicrobeNet [91]. Some institutions have also developed in-house spectral libraries, which have shown potential to further improve identification accuracy [92].

### 4.4. Molecular Methods for C. auris Detection

#### 4.4.1. Overview of DNA-Based Detection

DNA-based methods such as polymerase chain reaction (PCR) and loop-mediated isothermal amplification (LAMP) have proven highly effective for early *C. auris* detection. PCR typically produces results within 2 to 5 h, while LAMP assays can deliver detection within 30 min [87,93,94,95]. These tools are ideal for colonization screening and outbreak control due to their speed and operational ease.

However, limitations exist. PCR assays cannot differentiate between viable and non-viable cells, leading to possible overestimation of colonization, especially in non-sterile samples [13]. Similarly, LAMP’s performance is sensitive to sample integrity, with reduced sensitivity reported in cases of poor specimen handling or prolonged storage.

#### 4.4.2. PCR-Based Detection

Quantitative PCR (qPCR) assays for *C. auris* vary based on DNA extraction protocols, primer design, and amplification chemistry. Commonly targeted genetic regions include the internal transcribed spacer (ITS1 and ITS2); 5.8S, 18S, and 28S ribosomal DNA (rDNA); and the glycosylphosphatidylinositol (GPI) gene [94,96,97,98]. Among these, the ITS2 region is the most frequently used due to its multiple genome copies and enhanced sensitivity. The CDC has developed specific primers targeting ITS2 for public health use [99]. While the GPI gene offers improved specificity, it remains underutilized in routine diagnostics [97,98]. More recently, the NADH dehydrogenase subunit 5 (nad5) gene has been explored as an alternative target with promising specificity [100].

A comprehensive 2025 review of 28 studies identified eight commercial PCR assays and 16 laboratory-developed tests (LDTs) targeting *C. auris*. Commercial platforms demonstrated high diagnostic performance, with sensitivity ranging from 94.9% to 100% and specificity from 98.2% to 100%, while LDTs showed similar sensitivity (93.3–100%) and specificity (92–100%) [23,101].

In terms of regulatory approvals, the GenMark ePlex BCID-FP and BioFire FilmArray BCID2 panels have received FDA clearance for *C. auris* detection from blood cultures (510(k) Numbers: K182690, K200010) [102,103]. The DiaSorin Molecular Simplexa *C. auris* assay was granted FDA de novo authorization in July 2024 and validated for axilla/groin swab screening [104,105,106]. In Europe, three CE-IVD-marked PCR assays include the OLM AurisID, RealCycler *C. auris* PCR, and Eazyplex *C. auris* assays [23,40,95,107,108]. However, no FDA-approved high-throughput molecular platform currently exists for colonization screening, highlighting an urgent need for automation-compatible, scalable solutions.

#### 4.4.3. Loop-Mediated Isothermal Amplification (LAMP)

LAMP is an isothermal amplification technique performed at 60–65 °C using 4–6 primers and a strand-displacing DNA polymerase. It is praised for its simplicity, speed, and cost effectiveness. Commercial LAMP assays such as Eazyplex *C. auris* and LAMPAuris have demonstrated sensitivity between 66% and 86% and specificity from 96.8% to 100% across a range of clinical, surveillance, and environmental samples [95,109,110].

DH Lim et al. (2022) developed a multiplex pan-Candida LAMP assay using Chelex-100 DNA extraction, achieving 100% sensitivity and specificity in candidemia blood samples and effectively distinguishing six *Candida* species [110]. Nevertheless, LAMP performance is susceptible to sample quality. Studies have shown that improper storage or lack of pre-processing—such as brief vortexing—can significantly reduce sensitivity [95,109].

#### 4.4.4. T2 Magnetic Resonance (T2MR) Assay

The T2MR assay, originally developed for bloodstream infections, has been adapted for *C. auris* colonization screening using the fully automated T2DX system. This platform enables rapid (<3 h), highly sensitive detection at levels as low as 1–3 CFU/mL [24].

T2MR demonstrated 89% sensitivity and 98% specificity when applied to axilla/groin swab samples. This led to the development of the T2Cauris panel, which expands detection to include *C. auris*, *C. haemulonii*, *C. duobushaemulonii*, and *C. lusitaniae* in both blood and environmental samples [24]. However, the assay remains designated for research use only and currently lacks regulatory approval for clinical diagnostics.

#### 4.4.5. Summary for Molecular Testing

Molecular diagnostics offer rapid, culture-independent detection of *C. auris*, critical for outbreak management and colonization screening. Compared to culture-based methods, they provide faster turnaround and higher sensitivity, even in resource-limited settings. Real-time PCR remains the gold standard, delivering high sensitivity and specificity, while LAMP offers a faster, cost-effective alternative, though it is more susceptible to sample quality issues. T2MR provides highly sensitive results from blood and swab samples but is currently limited to research use. PCR should be prioritized for routine screening, with LAMP as a complementary option where PCR is unavailable. Despite advances, scalable, FDA-approved high-throughput assays for *C. auris* colonization detection are urgently needed to strengthen clinical and public health responses.

## 5. Genomic Typing and Outbreak Investigation in *C. auris*

Molecular typing plays a critical role in outbreak detection, strain surveillance, and understanding the transmission dynamics of *C. auris*. These methods help identify strain-specific characteristics, including patterns of antimicrobial resistance, virulence, and geographic dissemination, thereby informing clinical management and infection control strategies [74,111].

Among available methods, whole genome sequencing (WGS) is considered the gold standard for high-resolution phylogenetic analysis. WGS enables the differentiation of local transmission from new introductions by comparing single nucleotide polymorphisms (SNPs) across isolates [112]. To support standardized genomic surveillance, Welsh et al. developed an empirical benchmark dataset containing 23 *C. auris* genomes, which serves as a reference for evaluating phylogenomic pipelines [25]. Complementing this, the CDC maintains a public *C. auris* genomic surveillance project on the National Center for Biotechnology Information (NCBI) platform (accession no. PRJNA642852), along with various open-access analytical tools to support outbreak investigations [113].

Although WGS provides unparalleled resolution, its cost, turnaround time, and infrastructure requirements can limit its routine use. As a result, alternative genotyping methods have been developed to balance accuracy with feasibility. Short tandem repeat (STR) typing has emerged as a reliable alternative, offering high reproducibility and strong correlation with WGS findings. STR can effectively differentiate isolates that vary by more than 30 SNPs, making it suitable for both local and regional epidemiological surveillance [26,51,114].

Multilocus sequence typing (MLST), which targets four conserved loci (ITS, D1/D2, RPB1, and RPB2), is effective in distinguishing *C. auris* from phylogenetically related species. It is useful in tracking antifungal resistance patterns across clades but lacks the resolution required for investigating transmission within hospital outbreaks [27,115].

Amplified fragment length polymorphism (AFLP) has also been explored as a low-cost genotyping option. While it offers a rapid turnaround, its poor reproducibility and inconsistent concordance with WGS and STR data limit its reliability in high-stakes outbreak scenarios [28].

The IR Biotyper^®^, utilizing Fourier transform infrared (FTIR) spectroscopy, offers a rapid and cost-effective method for typing *C. auris* by analyzing cellular biochemical fingerprints. Studies have demonstrated its capability to cluster *C. auris* isolates into major geographic clades. However, its resolution is insufficient for distinguishing closely related outbreak strains, limiting its application in detailed hospital outbreak investigations [28,116,117].

In summary, genomic analyses consistently demonstrate that isolates involved in hospital outbreaks or clonal clades often exhibit minimal genomic variation, reinforcing the need for high-resolution tools for accurate differentiation [112,118]. While WGS remains the most precise method for outbreak tracking and global surveillance, STR typing offers a practical compromise between speed, cost, and phylogenetic resolution. In contrast, MLST and AFLP, while useful for species-level identification and broader resistance surveillance, are less suitable for resolving transmission events during hospital outbreaks [28]. FTIR spectroscopy provides a rapid, cost-effective method for clade-level typing of *C. auris*, but its limited resolution restricts both detailed outbreak analysis and antifungal resistance profiling.

## 6. Antifungal Susceptibility Testing (AFST) for *C. auris*

### 6.1. Multidrug Resistance in C. auris

*C. auris* presents significant therapeutic challenges due to its high rates of multidrug resistance. In the United States, approximately 90% of isolates exhibit resistance to fluconazole, 30% to amphotericin B (AMB), and under 2% to echinocandins [119]. The global emergence of pan-resistant strains underscores the urgent need for accurate antifungal susceptibility testing (AFST) to guide effective treatment and support infection control strategies [1,120].

### 6.2. Reference Methods and Interpretive Challenges

Broth microdilution (BMD) is recommended as the reference method for AFST by both the Clinical and Laboratory Standards Institute (CLSI) and the European Committee on Antimicrobial Susceptibility Testing (EUCAST). While tentative clinical breakpoints (CBPs) and epidemiologic cutoff values (ECVs) have been proposed, formal susceptibility thresholds for *C. auris* remain unestablished [121]. Interpretation of minimum inhibitory concentration (MIC) values is further complicated by inter-clade variability and regional differences in MIC distributions [63,122]. In practice, many laboratories follow the CDC’s tentative breakpoints, which are informed by MIC trends, animal model data, and known resistance markers [119].

One unique complication in *C. auris* testing is the Eagle effect, where isolates appear inhibited at lower caspofungin concentrations but demonstrate regrowth at higher levels. This phenomenon may result in inconsistent MIC readings and misclassification of susceptible isolates as resistant, particularly if additional echinocandins are not tested alongside caspofungin [123].

### 6.3. Performance of Commercial Testing Platforms

Commercial AFST systems—including VITEK 2, Sensititre YeastOne (SYO), MICRONAUT-AM, and gradient diffusion methods such as Etest and MIC Test Strip (MTS)—provide faster results than BMD but show variable concordance. Gradient diffusion tests offer a broader MIC range but are prone to subjective interpretation, especially near susceptibility thresholds. Among these, Etest generally aligns more closely with BMD than VITEK 2, although it tends to overestimate AMB resistance [124,125].

SYO may also overestimate AMB resistance and, due to the Eagle effect, inaccurately assess caspofungin susceptibility [126]. The MICRONAUT-AM platform has improved the accuracy of AMB susceptibility results but may underestimate fluconazole resistance [127]. In 2022, CLSI advised laboratories to interpret AMB MICs with caution due to variability across methods. Despite producing MIC values, these platforms lack definitive interpretive criteria, as no clinical trials have established susceptibility breakpoints for *C. auris*. This variability necessitates careful clinical interpretation of automated results.

### 6.4. Genotypic Resistance Mechanisms and Molecular Testing

Molecular studies have revealed key genetic mutations that confer antifungal resistance in *C. auris*. Resistance to fluconazole is commonly linked to mutations in the ERG11 gene, while ERG2 mutations affect amphotericin B susceptibility. Echinocandin resistance is associated with mutations in FKS1, and resistance to 5-fluorocytosine has been traced to alterations in FCY2, FCY1, and FUR1 genes [32,128]. Genotypic testing enables earlier detection of resistance profiles, allowing clinicians to make timely therapeutic adjustments before phenotypic test results become available [129].

### 6.5. MALDI-TOF MS-Based AFST: Emerging Innovations

MALDI-TOF MS is emerging as a novel tool for AFST. The Minimal Profile Change Concentration (MPCC) measures changes in the mass spectrum of microbial proteins in response to antifungal exposure. These changes are quantified using the composite correlation index (CCI), a statistical measure that compares protein spectra before and after antifungal exposure to detect drug-induced alterations. The CCI has demonstrated over 90% accuracy in identifying resistance to fluconazole and caspofungin, although its performance is more variable for anidulafungin [130,131].

Further innovations include the MALDI Biotyper Antibiotic Susceptibility Test Rapid Assay (MBT ASTRA), which has shown over 95% accuracy in identifying echinocandin resistance in *C. auris* [85]. While promising, these tools require additional validation and the establishment of clinical breakpoints before they can be reliably adopted into routine practice [132].

### 6.6. Summary for AFST

In summary, BMD remains the gold standard for *C. auris* susceptibility testing but requires improved clinical breakpoints for accurate interpretation. The Etest generally aligns well with the reference method, whereas commercial platforms exhibit variable reliability, necessitating caution when interpreting amphotericin B MICs. Genotypic assays offer faster detection of resistance markers but cannot replace phenotypic testing, which remains essential for confirmation. MALDI-TOF MS-based AFST shows promise but requires further clinical validation before routine use.

## 7. Conclusions

Effective management of *C. auris* depends on a layered diagnostic approach combining culture-based methods, species identification, molecular tools, and antifungal susceptibility testing. While culture is foundational, it should be paired with confirmatory methods due to limitations in speed and accuracy. MALDI-TOF MS enables rapid, reliable identification when supported by updated databases, and real-time PCR offers high-sensitivity, culture-independent detection—crucial for screening and outbreak control.

Genomic tools like WGS and STR typing enhance outbreak tracking and resistance surveillance, while broth microdilution remains the gold standard for susceptibility testing despite interpretive limitations. Diagnostic strategies should be adapted to local resources and epidemiology. To support implementation, we provide a recommended diagnostic workflow (Figure 1) and a comparative summary of tools (Table 1).

## Figures and Tables

**Figure 1 microorganisms-13-01461-f001:**
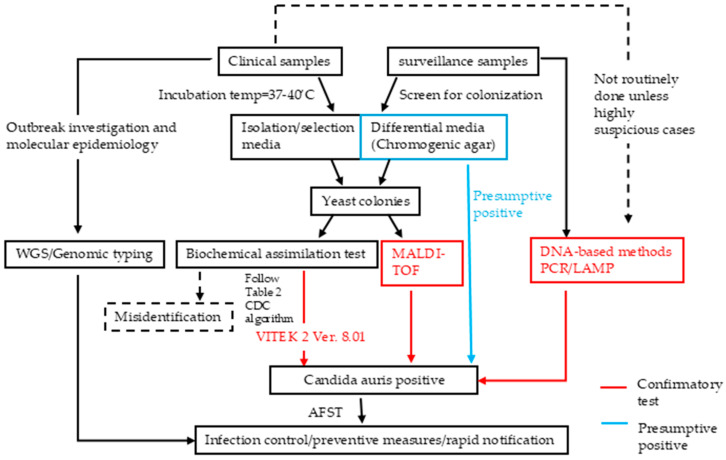
Workflow Algorithm for Detection in Clinical and Surveillance Samples. Clinical and surveillance samples are detected using culture-based methods, with colonies confirmed by MALDI-TOF and biochemical tests (e.g., VITEK 2). Biochemical tests risk misidentification and require CDC algorithm interpretation (Table 2). Alternative screening includes DNA-based confirmatory tests and chromogenic agar for presumptive identification. WGS aids outbreak and epidemiological investigations. Red lines indicate confirmatory tests. Abbreviations: AFST, antifungal susceptibility testing; BDG, beta-D-glucan; CDC, Centers for Disease Control and Prevention; MALDI-TOF MS, matrix-assisted laser desorption/ionization time-of-flight mass spectrometry; PCR, polymerase chain reaction; VITEK 2, VITEK 2 Biochemical Identification System; WGS, whole genome sequencing.

**Figure 2 microorganisms-13-01461-f002:**
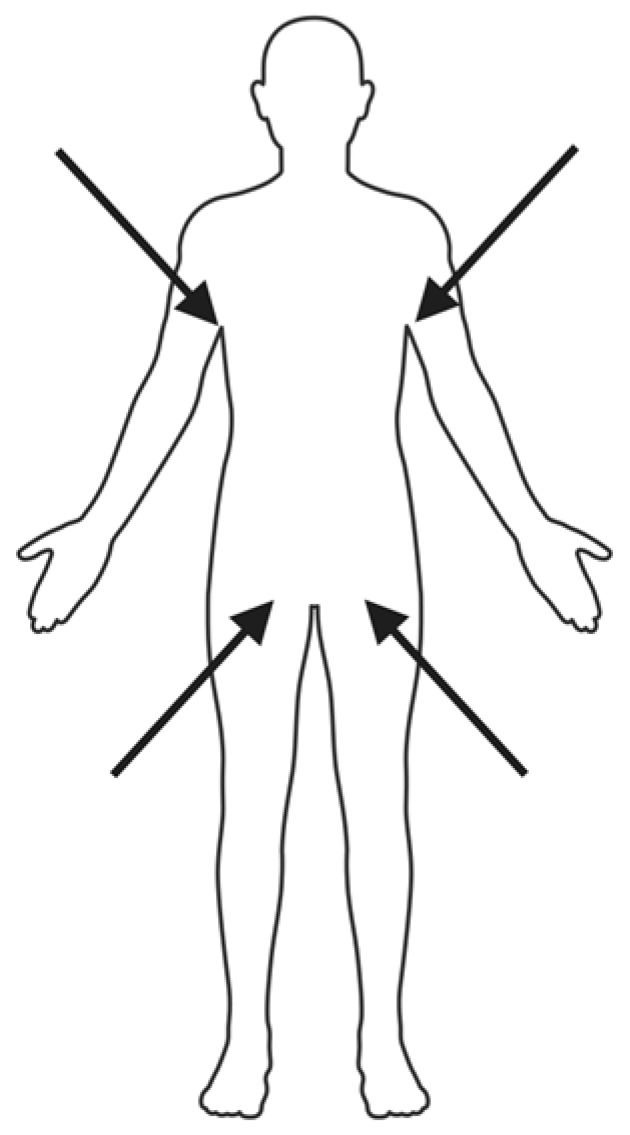
Recommended Sample Collection Sites for Screening. Swab the bilateral axillae and groins 3–5 times with the soft end of the collection swab. Recent studies suggest including nasal swabs to improve sensitivity.

**Figure 3 microorganisms-13-01461-f003:**
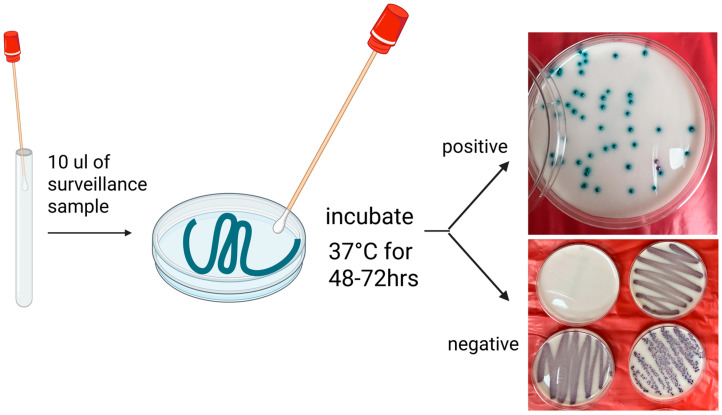
Workflow of Surveillance Sample Testing Using Chromogenic Agar. 10 µL of surveillance sample was streaked onto the culture media in a Z-pattern for optimal isolation. Plates were incubated at 37 °C for 48 to 72 h, with presumptive *C. auris* appearing with green colonies on the media (per manufacturer’s recommendation). Plates with no growth or colonies of other colors (e.g., white, pink, or purple) are considered negative.

**Figure 4 microorganisms-13-01461-f004:**
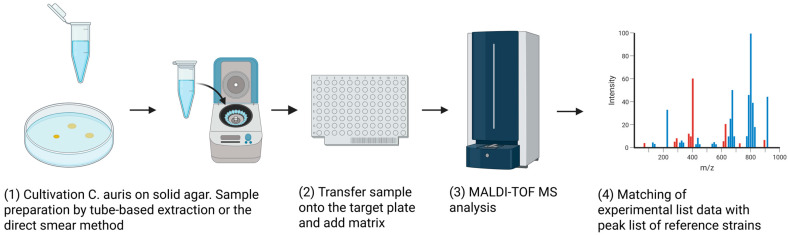
General workflow of matrix-assisted laser desorption ionization time-of-flight mass spectrometry (MALDI-ToF MS)-based identification analysis. (**1**) *C. auris* colonies are cultivated on solid agar, followed by sample preparation via tube-based protein extraction or direct smear methods. (**2**) Prepared samples are transferred onto a MALDI-TOF MS target plate and overlaid with a suitable matrix solution. (**3**) Samples are analyzed using MALDI-TOF MS to generate mass spectra. (**4**) Experimental mass spectra are matched against a database of reference strain spectra to achieve species-level identification, the colors represent different sample groups for comparison.

**Table 1 microorganisms-13-01461-t001:** Summary of Diagnostic tools for *C. auris*.

Category	Diagnostic Method	Mechanism	Highlights	Strengths	Weaknesses/Limitations	References
Culture-Based	Mycological media/Selective media/Chromogenic media	1. Requires higher incubation temperature(37–40 °C) than routine culture. 2. Adds saline, carbohydrates, or color indicators to differentiate from similar species.	1. SCA/SAM Media Se/Sp:100%/100%. 2. CHROMagarTM with Pal’s medium Se/Sp: 100%/100%. 3. CHROMagar™ Candida Plus Se/Sp: 90–100%/98–100%.	1. Basis for culture-based methods (e.g., biochemical assimilation methods and MALDI-TOF). 2. Simple, widely available, and low cost. 3. Gold standard for diagnosis in clinical samples; enables AFST.	1. Slow (48–72 h). 2. Various accuracy among different media; Non-specific colony morphology, prone to misidentification, and requires additional confirmation (MALDI-TOF MS or molecular methods).	[15,16,17,18,19,20]
Culture-Based	Biochemical Tests (VITEK, API, and Microscan)	Analysis of metabolic profile through carbohydrate assimilation, nitrogen utilization, and enzymatic activity.	1. Vitek 2 Version; 8.01 is confirmatory. 2. API 20C AUX or API ID 32C leads to frequent misidentification. 3. MicroScan/BD Pheonix has no *C. auris* database.	1. Low cost and easy to use (automated). 2. A rapid AFST tool reliable for azoles.	1. Requires database updates. 2. Various misidentification rates among different. commercial assays; often requires confirmatory test following CDC algorithm (Table 2).	[21]
Culture-Based	MALDI-TOF MS	Analysis of protein profiles and comparing them to reference databases (Figure 4).	1. Accuracy depends on databases, sample preparation, and instrument calibration. 2. Limited use for AFST with no clinical breakpoint of MPCC.	1. Rapid (4–5 h post culture). 2. Highly specific (Sp > 90%) when database is updated. 3. Cost effective.	1. Requires database updates. 2. Expensive equipment and limited access in resource-poor settings.	[22]
Culture-Independent	DNA-Based Assays (PCR/LAMP)	DNA amplification using species-specific primers (e.g., ITS and D1/D2 regions).	1. Both LDTs and commercial assays demonstrate reliable accuracy (Se/Sp > 90%); LAMP has lower sensitivity. 2. Current FDA-approved commercial assays for blood culture: GenMark ePlex BCID-FP and BioFire FilmArray BCID2.	1. Accurate and rapid, useful for colonization screening and outbreak control in healthcare settings.	1. Cannot determine antifungal susceptibility and detect DNA from both viable and non-viable *C. auris* cells. 2. LAMP has lower sensitivity.	[23]
Culture-Independent	T2 MR assay	Superparamagnetic nanoparticles bind target DNA/RNA, altering T2 relaxation for MR detection.		Rapid (<3 h) and highly sensitive detection for surveillance and bloodstream infection.	Research use only.	[24]
Culture-Independent	Whole Genome Sequencing (WGS)/genomic typing	High-resolution genetic analysis.	Tracking outbreaks, identifying strains, and analyzing resistance, virulence, and epidemiology; not usually used for individual diagnosis.	WGS: High resolution; gold standard for outbreaks investigation.	Costly, requires specialized analysis.	[25,26,27,28]
				STR typing: High reproducibility; aligns well with WGS.	Differentiate *C. auris* strains only if >30 SNP differences.	
				MLST: Differentiates *C. auris*; supports resistance surveillance.	Low resolution within clades; limited for outbreaks.	
				AFLP: Rapid, cost effective.	Poor reproducibility; inconsistent clustering.	
Culture-Independent	Beta-D-glucan (BDG) assays	Detection of fungal cell wall components		Non-invasive; useful for early detection for clinical samples.	Limited sensitivity and low specificity for invasive candidiasis.	[29,30]

Diagnostic tools for *C. auris* are broadly divided into culture-dependent and culture-independent methods; this table summarizes the mechanisms, updated highlights, strengths, and limitations of each tool. Abbreviations: Se, sensitivity; Sp, specificity; SCA, specific *C. auris* Medium; SAM, selective auris medium; CHROMagar™, chromogenic agar; AFST, antifungal susceptibility testing; VITEK 2, VITEK 2 Biochemical Identification System; API, analytical profile index; MALDI-TOF MS, matrix-assisted laser desorption/ionization time-of-flight mass spectrometry; MPCC, minimal profile change concentration; PCR, polymerase chain reaction; LAMP, loop-mediated isothermal amplification; FDA, U.S. Food and Drug Administration; T2 MR, T2 magnetic resonance; WGS, whole genome sequencing; STR, short tandem repeat; SNP, single-nucleotide polymorphism; MLST, multilocus sequence typing; AFLP, amplified fragment length polymorphism; BDG, beta-D-glucan.

**Table 2 microorganisms-13-01461-t002:** CDC Guidelines for Identifying *C. auris* Using Biochemical Tests.

Identification Method	Database/Software, If Applicable	Is Confirmed If Initial Identification Is *C. auris.*	Is Possible If the Following Initial Identifications Are Given; Further Work-Up Is Needed to Determine If the Isolate Is *C. auris*
Bruker Biotyper MALDI-TOF	RUO libraries (Versions 2014 [5627] and more recent)		n/a
	CA System library (Version Claim 4)		n/a
bioMérieux VITEK MS MALDI- TOF	RUO library (with Saccharomycetaceae update)		n/a
	IVD library (v3.2)		n/a
	Older IVD libraries	n/a	*C. haemulonii* *C. lusitaniae* No identification
VITEK 2 YST	Software version 8.01		*C. haemulonii* *C. duobushaemulonii*; *Candida* spp. not identified
	Older versions	n/a	*C. haemulonii* *C. duobushaemulonii*; *Candida* spp. not identified
API 20C		n/a	*Rhodotorula glutinis* (without characteristic red color) *C. sake* *Candida* spp. not identified
API ID 32C		n/a	*C. intermedia* *C. sake* *Saccharomyces kluyveri*
BD Phoenix		n/a	*C. catenulata* *C. haemulonii* *Candida* spp. not identified
MicroScan		n/a	*C. lusitaniae* *C. guilliermondii* *C. parapsilosis* *C. famata* *Candida* spp. not identified
RapID Yeast Plus		n/a	*C. parapsilosis* *Candida* spp. not identified
GenMark ePlex BCID-FP Panel			n/a

CDC guidelines for confirming or suspecting *C. auris* based on biochemical assimilation tests and database versions. Abbreviations: RUO, research use only; CA System, chromogenic agar system; IVD, in vitro diagnostic; VITEK 2 YST, VITEK 2 Yeast Susceptibility Testing; API 20C AUX, Analytical Profile Index 20C Auxanographic Yeast Identification System; API ID 32C, Analytical Profile Index ID 32C Yeast Identification System; BD Phoenix, Becton Dickinson Phoenix Identification System; MicroScan, MicroScan WalkAway Identification System; RapID, RapID Yeast Plus System; GenMark ePlex BCID-FP, GenMark ePlex Blood Culture Identification–Fungal Pathogen Panel.

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
