# Peer review of "Diagnostic Approaches for Candida auris: A Comprehensive Review of Screening, Identification, and Susceptibility Testing"

_microorganisms, 2025, doi:10.3390/microorganisms13071461_

Round 1
Reviewer 1 Report
Comments and Suggestions for Authors
The authors made a comprehensive and up-to-date review of all diagnostic methods for the detection of both infection and colonization with C.auris. There is a summary for each group of methods, underscoring the benefits and the limits of each.
There are few typos/missing words on lines 112, 133 (n the title), 446 and 562
More precise explanations for the negative plates in Figure 3 would be beneficial.
Please reformulate/rearranged the table head for Tabel number 2 (below line 321). It seems a bit confusing for the reader.
Author Response
Response to Reviewer 1’s Comments
We sincerely thank reviewer 1 for the thoughtful review and constructive feedback on our manuscript. Please find below our point-by-point responses:
Comment 1:
“The authors made a comprehensive and up-to-date review of all diagnostic methods for the detection of both infection and colonization with C. auris. There is a summary for each group of methods, underscoring the benefits and the limits of each.”
Response:
We appreciate the positive feedback and are grateful for the recognition of the scope and structure of our review.
Comment 2:
“There are few typos/missing words on lines 112, 133 (in the title), 446 and 562.”
Response:
Thank you for pointing this out. We have carefully reviewed and corrected the typographical errors and missing words on the mentioned lines, including the title at line 133.
Comment 3:
“More precise explanations for the negative plates in Figure 3 would be beneficial.”
Response:
We have revised the figure legend and corresponding text in the manuscript to provide a clearer explanation. Specifically, we now state: “Plates with no growth or colonies of other colors (e.g., white, pink, or purple) are considered negative for C. auris.”
Comment 4:
“Please reformulate/rearrange the table head for Table number 2 (below line 321). It seems a bit confusing for the reader.”
Response:
We have reformatted and clarified the header of Table 2 to enhance readability. Additionally, we have renamed the table to:
“Table 2. CDC Guidelines for Identifying C. auris Using Biochemical Tests.”
We hope these revisions fully address the concerns raised and enhance the clarity and quality of the manuscript. Thank you again for your valuable feedback and support.
Sincerely,
Christine Hsu, Mohamed Yassin
Reviewer 2 Report
Comments and Suggestions for Authors
In this study, the authors review C. auris in terms of laboratory techniques for culturing, isolation, and differentiation. Also, identification challenges and screening practices are assessed. The role of MALDI-TOF MS technique and molecular methods especially PCR and LAMP, as well as T2MR, are also presented. Tools in investigating outbreaks are evaluated. Available methods on antifungal susceptibility testing are presented and their drawbacks are discussed. The manuscript is very well written and references cover most aspects.
Some minor points:
Line 125-127: beta-D-glucan has lower sensitivity especially for C. auris and C. parapsilosis. authors could denote this and add a reference.
Line 133: “of…” please complete the sentence or delete “of”
Line 272-3 and 303: C. haemulonii should be in italics
Line 456: complete the title of this section or delete “in”
Line 456-494: Another technology that has been used in few cases in investigating outbreaks is Fourier transform infrared spectroscopy (FT-IR) technology. The authors could refer briefly to this option.
Line 554-560: May the authors denote for one more time the overestimation of amphotericin B resistance by commercial tests.
Line 562: “of….” please complete the sentence
Author Response
Response to Reviewer 2’s Comments
We thank Reviewer 2 for their careful reading of the manuscript and for the positive and constructive feedback. Please find our detailed responses below:
Comment 1:
“Line 125–127: β-D-glucan has lower sensitivity especially for C. auris and C. parapsilosis. Authors could denote this and add a reference.”
Response:
Thank you for the suggestion. We have added a sentence to highlight the lower sensitivity of β-D-glucan assays for C. auris and C. parapsilosis, along with appropriate references ([33, 34]) in lines 127–128.
Comment 2:
“Line 133: ‘of…’ please complete the sentence or delete ‘of’.”
Response:
We have revised the sentence at line 133 by completing it and adding "C. auris" after “of” for grammatical accuracy.
Comment 3:
“Lines 272–273 and 303: C. haemulonii should be in italics.”
Response:
We have corrected the formatting and italicized C. haemulonii at both locations as requested.
Comment 4:
“Line 456: complete the title of this section or delete ‘in’.”
Response:
We have revised the section title at line 456 by adding "C. auris" after “in” to ensure the heading is complete and clear.
Comment 5:
“Lines 456–494: Another technology that has been used in few cases in investigating outbreaks is Fourier transform infrared spectroscopy (FT-IR) technology. The authors could refer briefly to this option.”
Response:
We appreciate this valuable suggestion. We have now added a brief mention of FT-IR technology as an emerging method for outbreak investigations, along with relevant references ([28, 116, 117]) in lines 488–493. A short summary of its potential applications has also been included in lines 501–503.
Comment 6:
“Lines 554–560: May the authors denote for one more time the overestimation of amphotericin B resistance by commercial tests.”
Response:
As suggested, we have added a sentence in line 568 to emphasize the need for caution when interpreting amphotericin B resistance results obtained from commercial susceptibility testing methods.
Comment 7:
“Line 562: ‘of…’ please complete the sentence.”
Response:
We have completed the sentence at line 562 by adding "C. auris" after “of” to ensure clarity and remove the fragment.
We thank the reviewer once again for their insightful comments, which have helped improve the clarity and scientific accuracy of the manuscript.
Sincerely,
Christine Hsu, Mohamed Yassin
Reviewer 3 Report
Comments and Suggestions for Authors
The authors of the manuscript "Diagnostic Approaches for Candida auris: A Comprehensive Review of Screening, Identification, and Susceptibility Testing" comprehensively described the diagnostic protocol of the drug-resistant pathogen Candidia auris. Manuscript well edited correctly described procedures for identifying pathogenic microorganism on both clinical and control trials. The only remarks that could improve the clarity of the written text are related to the extension of the manuscript by two short chapters. In the first one, the authors should describe the symptoms of infection in more detail, but not only briefly describe the places of the cleanest infection. In the second chapter, I briefly write about the methods of treatment.
Author Response
Response to Reviewer 3’s Comments
We sincerely thank Reviewer 3 for their thoughtful review and kind remarks regarding the quality, organization, and accuracy of our manuscript. We appreciate your positive feedback on the clarity of the text and the comprehensive coverage of diagnostic procedures for Candida auris. Please find our response to your suggestion below:
Comment:
“The only remarks that could improve the clarity of the written text are related to the extension of the manuscript by two short chapters. In the first one, the authors should describe the symptoms of infection in more detail, but not only briefly describe the places of the cleanest infection. In the second chapter, I briefly write about the methods of treatment.”
Response:
We thank the reviewer for these thoughtful suggestions. While we agree that the clinical presentation and treatment options for Candida auris are important topics, we respectfully note that the primary aim of our manuscript is to provide a focused and up-to-date review of diagnostic tools and laboratory methods used to detect and identify C. auris, including screening strategies, molecular techniques, and susceptibility testing.
Expanding the manuscript to include sections on symptoms and treatment would shift the scope beyond our intended emphasis on diagnostics. Moreover, these topics are already well-covered in other recent reviews and clinical guidelines.
Therefore, we believe maintaining the current focus on diagnostic approaches best serves the purpose and clarity of the manuscript. That said, we have made a minor addition in the introduction to briefly acknowledge the relevance of clinical manifestations and treatment options, with appropriate references, to provide context while keeping the diagnostic emphasis intact.
We once again thank the reviewer for the helpful input and hope this explanation clarifies our decision to retain the current scope of the manuscript.
Sincerely,
Christine Hsu, Mohamed Yassin